# Learning Unknowns from Unknowns: Diversified Negative Prototypes Generator for Few-Shot Open-Set Recognition

## ABSTRACT

Few-shot open-set recognition (FSOR) is a challenging task that requires a model to recognize known classes and identify unknown classes with limited labeled data. Existing approaches, particularly Negative-Prototype-Based methods, generate negative prototypes based solely on known class data. However, as the unknown space is infinite while the known-space is limited, these methods suffer from limited representation capability. To address this limitation, we propose a novel approach, termed **D**iversified **N**egative **P**rototypes **G**enerator (DNPG), which adopts the principle of "learning unknowns from unknowns." Our method leverages the unknown space information learned from base classes to generate more representative negative prototypes for novel classes. During the pre-training phase, we learn the unknown space representation of the base classes. This representation, along with inter-class relationships, is then utilized in the meta-learning process to construct negative prototypes for novel classes. To prevent prototype collapse and ensure adaptability to varying data compositions, we introduce the Swap Alignment (SA) module. Our DNPG model, by learning from the unknown space, generates negative prototypes that cover a broader unknown space, thereby achieving state-of-the-art performance on three standard FSOR datasets. We provide the source code in the supplementary materials for reproducibility.

## CCS CONCEPTS

• **Computing methodologies** → **Computer vision**.

## KEYWORDS

Few-shot, Open-set, Learning unknowns from unknowns, Negative-Prototype-Based, Diversified negative prototypes

## 1 INTRODUCTION

Deep learning [9, 13, 14] has significantly advanced various domains, largely due to its ability to leverage extensive training data. Conventionally, deep models are trained under the closed-set assumption, where the classes in the training data align with those in the test data. However, real-world scenarios often present more complex challenges. Firstly, acquiring a large volume of labeled data is frequently impractical, especially when data collection is costly or involves sensitive information. For instance, datasets for

*ACM MM, 2024, Melbourne, Australia*
© 2024 Copyright held by the owner/author(s). Publication rights licensed to ACM.
ACM ISBN 978-x-xxxx-xxxx-x/YY/MM
https://doi.org/10.1145/nnnnnnn.nnnnnnn

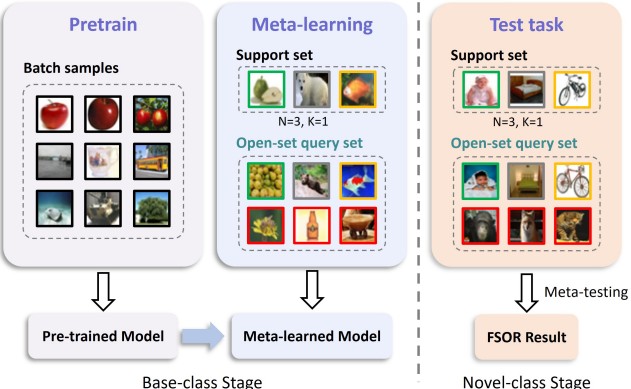

Figure 1: Task setting for FSOR: The model undergoes a two-phase pre-training on base class data, followed by testing where it classifies novel classes (green, gray, orange boxes) and identifies unknown class samples (red box).

rare disease diagnosis are typically small, and new users in recommendation systems have limited records. Secondly, models must contend with unknown data that falls outside the scope of training classes, such as online shopping platforms needing to identify and update new user-uploaded products.

To address these challenges, Few-Shot Open-Set Recognition (FSOR) [10, 15, 17, 22, 31, 35] has emerged as a critical task. FSOR demands that a model utilize minimal training data from the support set to effectively recognize the query set, which involves classifying known-class samples and identifying unknown-class samples. This can be viewed as an N+1 classification problem (add an "Unknown" class), as depicted in Figure 1. The FSOR model undergoes pre-training on base classes with ample data, which do not overlap with novel classes, before transitioning to the novel-class stage.

Since the unknown class can be understood as an extra class against known classes, a common strategy is to learn a representation for this unknown class, referred to as the negative prototype (NP), and perform an N+1 classification task using the prototypes of both known and unknown classes [6]. Existing methods [15, 31, 41] typically generate NPs based on known-class data, as depicted in Figure 2(a). However, this approach inherently limits the diversity of the NPs, as they tend to retain characteristics of the known classes. For instance, an NP generated for the "Dalmatian" class might resemble a generic dog with altered features, failing to represent the vast diversity of the unknown space. The unknown space is theoretically infinite, encompassing a wide range of samples that cannot be directly derived from known classes, such as a guitar being an unknown sample relative to the "Dalmatian" class.

To overcome this limitation, we propose a novel approach to generate diversified NPs by **learning unknowns from unknowns**, thus alleviating the constraints imposed by known-class information. As illustrated in Figure 2(b), during the base-class stage, we

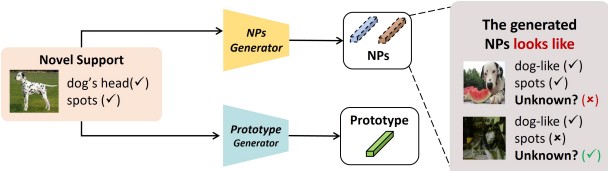

(a) Existing Negative-Prototype-Based FSOR method.

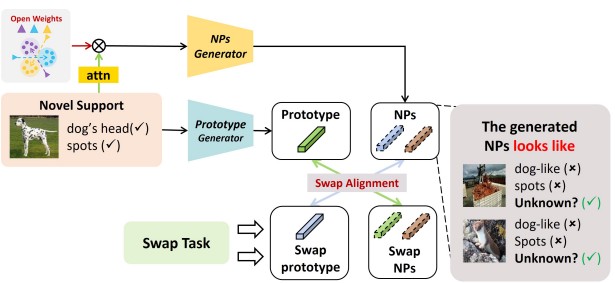

(b) Our Diversified Negative Prototypes Generator method.

**Figure 2: (a) Existing SOTA FSOR methods, like ATTG [15], generate Negative Prototypes (NPs) from support samples, leading to limited diversity (e.g., "Dalmatian" NPs resemble dog-like animals). (b) Our DNPG model leverages the unknown space representation (open weights) of base classes to produce diversified NPs in the test phase.**

leverage extensive data to learn the inverse representation of each base class, which we term the *open weight*. This representation captures the essence of **what is not that class**, effectively representing the corresponding unknown space. During the novel-class stage, utilizing these open weights to generate NPs allows for a broader coverage of the unknown space, enhancing the representation of diverse unknown-class samples. For example, open weights derived from base classes such as cars and planes could correspond to entities like rocks, trees, or the sun, which are unrelated to the given base classes. Consequently, during the novel-class stage, the model can generate diversified NPs for the "Dalmatian" class that are not confined to dog-like animals as shown in Figure 2(b).

However, a challenge arises as the open weights, derived from base classes, remain static during the novel-class stage. This can lead to the generated NPs collapsing into a single point, reducing their effectiveness. To address this, we further introduce the Swap Alignment (SA) module, which ensures that NPs model distinct unknown spaces for different novel-class data, thereby preserving their diversity. During the base-class stage, we sample a set of non-overlapping base classes as pseudo-unknown classes and minimize the distance between the generated NPs and these pseudo-unknown classes. Simultaneously, known class samples and pseudo-unknown class samples together form the swap tasks, where, at the prototype level, the distances between NPs and the pseudo-unknown class samples are further reduced. This approach ensures that each generated NP is distinct from all current known-class representations, preventing collapse and enabling adaptation to the current known classes.

In summary, our main contributions are as follows.

- We propose a novel FSOR method, the Diversified Negative Prototypes Generator (DNPG), to generate diversified negative prototypes by learning unknowns from unknowns, using the inverse representation of base classes.
- We introduce the Swap Alignment module to prevent NP collapse and enhance adaptation to novel-class data, further increasing the diversity of NPs.
- Extensive experiments have been carried out on multiple widely used FSOR datasets to prove that our DNPG model is superior to the current state-of-the-art methods.

## 2 RELATED WORK

**Few-Shot Learning.** Few-shot learning in visual classification addresses the challenge of classifying with limited support samples. Methods can be broadly categorized into transfer learning, which leverages pre-trained models from related tasks [33, 37], and meta-learning, which includes model-based [4, 23], metric-based [12, 30], and optimization-based approaches [1, 16]. Model-based methods adapt quickly to new tasks with minimal data, metric-based methods optimize the distance distribution between samples, and optimization-based methods focus on efficient model training with limited examples.

**Open-Set Recognition.** Current open-set recognition methods can be categorized based on their approach to discriminating unknown class samples, including similarity threshold-based methods [2, 8, 29], and density threshold-based methods [20, 39]. Additionally, a new trend has emerged, where Gaussian Mixture Models (GMMs) are applied to model closed-set distributions [7, 21], enabling the models to directly reject unknown class samples. However, no previous work has employed GMMs to address the FSOR task. This is mainly attributed to the difficulty of learning the joint distribution of classes and pixel features when only a limited number of labeled samples from the novel classes are available.

**Few-Shot Open-set Recognition.** FSOR combine the challenges of FSL and OSR, requiring the completion of OSR tasks with limited support samples. PEELER [22] proposes to train and test FSOR tasks in a meta-learning pattern. RFDNet [10] suggests meta-learning a feature displacement relative to a pre-trained reference feature embedding. SnaTCHer [17] identifies unknown samples based on the transformation consistency, which measures the difference between transformed prototypes and a modified prototype set. The GEL [35] model utilizes an energy function as the discriminative criterion and introduces a pixel-level prototype network. ReFOCS [24] adopts a generative approach to dynamically adjust the similarity threshold by reconstructing samples. OPP [32] proposes constructing an overall positive prototype and then filters out unknown class samples using a threshold method. MSCL [40] combines the strengths of supervised contrastive learning and meta-learning to effectively increase inter-class distinctions and reinforce intra-class compactness. MRM [5] proposes enlarging the margin between different classes by extracting the multi-relationship of paired samples to dynamically refine the decision boundary for known classes and implicitly delineate the distribution of unknowns.

**Negative-Prototype-Based FSOR Methods.** These FSOR methods learns one or multiple prototypes for unknown class samples,

thereby transforming the open-set recognition task into an N+1 classification task. Existing Negative-Prototype-Based FSOR methods: ProCAM [31] learns NPs by extracting the background of support set images, ATTG [15] uses attention mechanisms to generate NPs that are distant from known class samples. ASOP [18] proposes to generate a negative prototype closest to open-set samples and second-closest to closed-set samples. TNPNet [38] adopts a transductive learning approach, leveraging both local and global contexts to enhance prototypes and negative prototypes. However, these methods utilize known class samples from the support set to learn NPs, which inevitably leads to the NPs inheriting the characteristics of the known class samples. As a consequence, the learned NPs are limited in their ability to model the unknown space, as illustrated in Figure 2 (a).

**Multiple Prototypes Model.** This approach involves generating multiple prototypes to represent a batch of data. Methods like [6, 15] generate multiple negative prototypes, but often face challenges such as prototype collapse and ineffective utilization of prototypes. Some studies [25, 36] attempt to address these issues with an equipartition constraint, but these methods may not perform well in small-sample open-set recognition tasks and could potentially decrease model performance.

## 3 PRELIMINARIES

### 3.1 Few-Shot Open-Set Recognition Setting

FSOR task aims to achieve accurate classification of known classes with limited support samples while effectively identifying instances from unknown classes. Formally, an FSOR task $\mathcal{T}$ is defined as a triplet $\mathcal{T} = \{\mathcal{S}, Q_k, Q_u\}$, comprising a support set $\mathcal{S} = \{x_i, y_i\}_{i=1}^{|S|}$, a query set $Q_k = \{x_i, y_i\}_{i=1}^{|Q_k|}$ for known classes, and an open-set query set $Q_u = \{x_i, y_i\}_{i=1}^{|Q_u|}$ for unknown classes. The support set $\mathcal{S}$ contains $K$ samples per class, where $K$ is typically small (e.g., 1 or 5). The labels $y$ for the support set $\mathcal{S}$ and query set $Q_k$ belong to the set of novel known classes $C_k$, while the labels for $Q_u$ belong to the set of unknown classes $C_u$, with $C_k \cap C_u = \emptyset$.

A proficient FSOR model should: (1) accurately classify images from the query set $Q_k$ into the corresponding known classes using the knowledge encapsulated in the support set $\mathcal{S}$, and (2) effectively recognize images from classes not represented in the current support set (i.e., samples from $Q_u$) as belonging to unknown classes.

### 3.2 Negative-Prototype-Based FSOR

Negative-prototype-based FSOR approaches transform the open-set recognition problem into an $(N + 1)$-way closed-set classification task by introducing Negative Prototypes (NPs). In the absence of unknown-class samples, a prototype-based few-shot classifier can be formulated as

$$\text{argmax}_c \left( \{f_s \left(\mathcal{F}_\theta(q), \mathbf{p}_c\right)\}_{c \in C_k} \right), \tag{1}$$

where $\mathcal{F}_\theta$ is the feature extractor, $f_s$ is the similarity fynction, $C_k$ denotes the set of novel known classes, and $\mathbf{p}_c$ represents the prototype for class $c$, computed as the average feature vector of samples belonging to class $c$. To accommodate unknown-class samples, Negative-Prototype-Based FSOR models extend the prototype-based classification by learning an additional NP for the unknown

classes. The classification then involves $(|C_k| + 1)$ prototypes:

$$\text{argmax}_c \left( \{f_s \left(\mathcal{F}_\theta(q), \mathbf{p}_c\right)\}_{c \in C_k \cup \{\text{unknown}\}} \right), \tag{2}$$

where a sample is classified as belonging to an unknown class if it is most similar to the NP.

However, as illustrated in Figure 2(a), existing approaches often derive NPs based on support samples, inadvertently incorporating information from known classes. This can lead to NPs that closely resemble known classes or highly similar classes (e.g., Dalmatians and Alaskans), which complicates the modeling of unknown classes within the diverse space of unknowns.

## 4 THE PROPOSED METHOD

In this section, we introduce our approach to tackle the FSOR problem by generating diverse NPs. Our model architecture is depicted in Figure 3. We first describe our baseline model, followed by our strategy for learning open-set weights for base classes. Finally, we present our Diversified Multiple Negative Prototype Generator (DMNPG) module and the Swap Alignment module, which together facilitate the generation of task-level diversified NPs for different tasks.

### 4.1 Baseline Model

Our baseline model is built upon the framework proposed by Huang et al. [15] and utilizes a metric-based meta-learning architecture, similar to previous FSOR methods [12, 15, 30]. As shown in Figure 1, the training process is divided into two stages: the base-class stage, which uses a large-scale dataset of base classes, and the novel-class stage, which evaluates the model on a dataset of novel classes with all model parameters fixed.

**Pre-training.** During pre-training, we employ a ResNet-12 network to perform a large-scale classification task using labeled samples from the base classes. The resulting feature extractor $\mathcal{F}_\theta$ and the classifier head for base classes are obtained, with the classifier head weights denoted as $P^*$.

**Meta-learning.** In the meta-learning stage, the base class dataset is redivided into different tasks. Sampling 2N classes in each task, with N known classes ($\mathcal{S}$ and $Q_k$) and N designated as pseudo-unknown classes ($Q_u$). The pre-trained Res-12 network serves as the feature extractor $\mathcal{F}_\theta$ to extract low-dimensional feature representations $v_i$ for samples in each task. For each task, the feature representations in the support set are averaged by class to obtain the raw class prototype $p_c^r$:

$$p_c^r = \frac{1}{K} \sum_{i=1}^{K} \mathcal{F}_\theta \left(x_{c,i}^{\mathcal{S}}\right), \quad v_i = \mathcal{F}_\theta \left(x_i\right), \tag{3}$$

where $x_{c,i}^{\mathcal{S}}$ represents the $i$-th sample of class $c$ from the support set, and $p_c^r \in P^r$ is the raw class prototype of class $c$, with $c \in C_S$. A Multi-Layer Perceptron (MLP) is then applied to the averaged class prototype $p_{avg}^-$ to generate the NP:

$$p^- = f^{mlp} \left(p_{avg}^-\right), \quad p_{avg}^- = \frac{1}{|C_k|} \sum_{c \in C_k} p_c^r, \tag{4}$$

where $p_{avg}^-$ is the average of the raw class prototypes, considering all novel classes in the current task to generate a task-specific NP.

After obtaining both the raw class prototypes $P^r$ and the NP $p^-$, the closed-set classification and open-set recognition tasks are

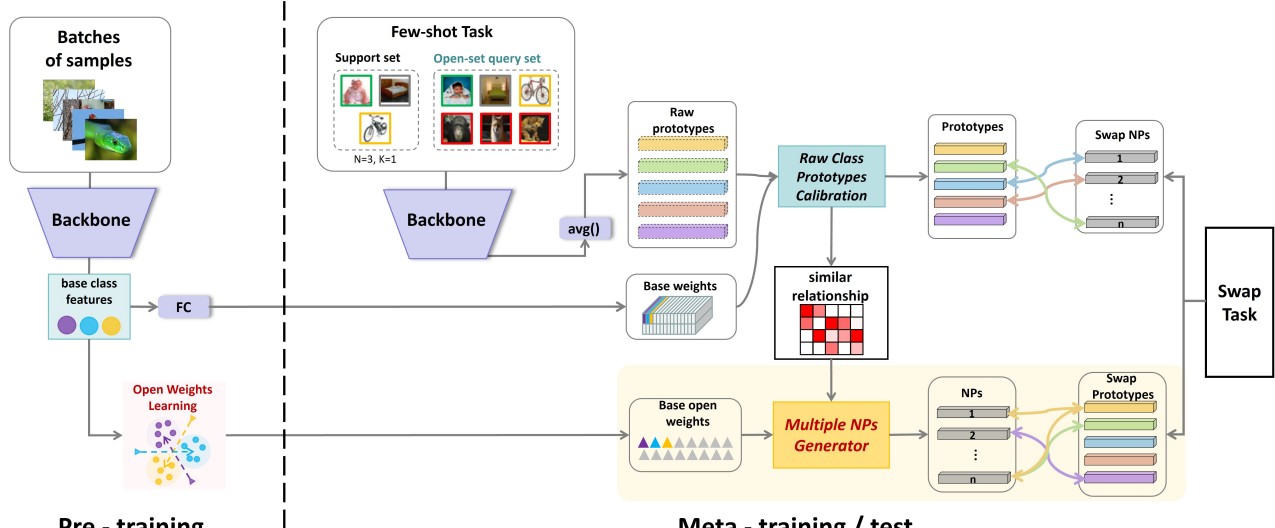

**Figure 3: An overview of the base-class stage in our DNPG model. In the pre-training phase, the model learns an inverse representation, termed as open weight, for each base class. During the meta-learning phase, these open weights, along with the similarity relationship between base and novel classes, are utilized to generate Novel Prototypes (NPs). Furthermore, the Swap Alignment module is employed to guide the NP generation process, thereby improving their diversity.**

combined into a $(|C_k|+1)$-way classification task:

$$\arg\max_c \left( \{ f_s \left( \mathcal{F}_\theta \left( q \right), [P^r, p^-] \right) \}_{c \in C_k \cup unknown} \right), \quad (5)$$

where $f_s$ is a function to measure the similarity between two inputs, e.g., cosine similarity. The cross-entropy loss is used to train the $(|C_k|+1)$-way classification baseline model as $\mathcal{L}_{CE} \left( Q_k \cup Q_u \right)$.

**Testing.** During the testing phase, the model parameters are fixed. For a test task $\mathcal{T} = \{S, Q_k, Q_u\}$, prototypes of novel classes and the NP are generated using the support set samples $S$, as per the equations above. The cosine similarity between the query samples $\{Q_k, Q_u\}$ and the prototypes, including the NP, is calculated. A query sample is classified into the corresponding novel class if its cosine similarity with a novel class prototype is the highest. However, if its cosine similarity with the NP is the highest, it is considered to be from an unknown class.

## 4.2 Diversified multiple NPs generator

Our method aims to generate negative prototypes directly from the unknown space by utilizing the base-class inverse representation, referred to as **open weights**, and leveraging the similarity relationships between classes.

**Learning the transferable open weights.** In the pre-training phase, we adopt the concept of Reciprocal Prototype Learning (RPL) [6] to learn the open weight $O_j$ for each base class $j$. The objective is to position $O_j$ far from the samples of class $j$ while maintaining proximity to samples from other base classes. This is achieved through a reverse classification task where the probability of a sample being classified as class $j$ is inversely proportional to its distance from $O_j$:

$$p \left( y = j \mid x, \mathcal{F}_\theta, O_j \right) = \frac{e^{d(\mathcal{F}_\theta(x), O_j)}}{\sum_{k=1}^N e^{d(\mathcal{F}_\theta(x), O_j)}}. \quad (6)$$

After obtaining the classification probability, the cross entropy loss function is used to train the model:

$$\mathcal{L}_b^- \left( x; \theta, O_j \right) = -\log p \left( y = k \mid x, \mathcal{F}_\theta, O_j \right), \quad (7)$$

where $d$ is the Euclidean distance function. The learned open weight $O_j$ characterizes features outside class $j$'s domain, representing its unknown space. The matrix of all base classes' open weights is denoted as $O$. To reduce training overhead and improve DNPG model generalization, we fix the feature extractor during open weights learning, allowing only the open weights to be trainable.

**Raw Class Prototypes Calibration** Inspired by [11], we refine the raw class prototypes $P^r$ for novel classes during the meta-learning phase using the base class weights $P^*$ (refer to Figure 4, left). Specifically, we adjust $P^r$ to incorporate characteristics akin to $P^*$, resulting in the final class prototypes $P$:

$$\mathbf{A}_{(Pr, P*)} = \frac{1}{\sqrt{d}} \left( P^r W^q \left( P^* W^k \right)^T \right), \quad (8)$$

$$P = P^r + \sigma \left( \mathbf{A}_{(Pr, P*)} \right) \left( P^* W^v \right), \quad (9)$$

where $W^q, W^k, W^v \in \mathbb{R}^{d \times d}$ are trainable parameters, and $\sigma$ denotes the softmax operation. The resulting weight matrix $\mathbf{A}_{(Pr, P*)} \in \mathbb{R}^{|C_S| \times |C_B|}$ captures the learned relationships between novel and base classes for task $T$.

**Multiple NPs Generator Based on Open Weights** The open weights, representing the unknown space of base classes, reveal that samples from similar classes exhibit analogous characteristics in this space (as illustrated in Figure 5). Exploiting this observation, we generate Negative Prototypes (NPs) for novel classes using the open weights of base classes (see Figure 4, right).

Specifically, leveraging the class similarity matrix $\mathbf{A}_{(Pr, P*)}$ from Eq. 8, we construct the NPs by allowing novel classes to mimic the

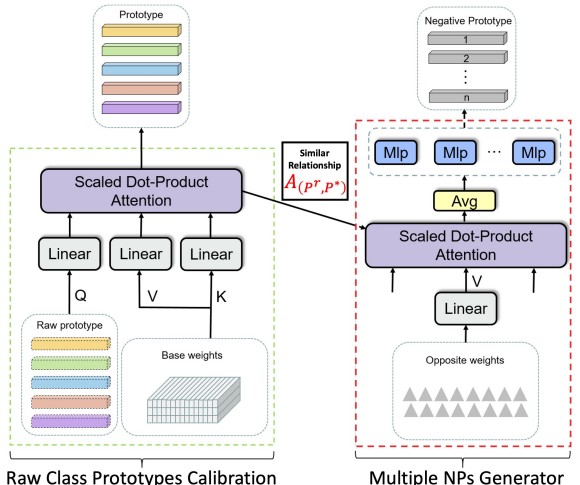

**Figure 4: Details of the Raw Class Prototypes Calibration (RPC) and Multiple NPs Generator (MNG) modules. In the RPC module, a standard Transformer attention block is utilized to calibrate raw prototypes with base weights. Subsequently, in the MNG module, based on the similarity relationship established in the RPC module, open weights are employed to generate multiple NPs for the current episode.**

unknown space characteristics of base classes, while preserving the class similarity relationships. The NPs for each novel class are generated as follows:

$$P^- = \sigma\left(\mathbf{A}_{(P^r, P^*)}\right)\left(OW^v\right), \tag{10}$$

where $W^v$ is a trainable parameter, and $O$ and $\mathbf{A}_{(P^r, P^*)}$ represent the open weights of base classes. Each NP, $P_j^- \in P^-$, signifies the negation of novel class $j$. To optimize $P_j^-$, we employ a binary cross-entropy loss that minimizes its similarity with queries of class $j$ and maximizes its similarity with queries from other classes:

$$\mathcal{L}_n^{neg}(j) = \frac{1}{\left|Q_k^j\right|} \sum_{q \in Q_k^j} \mathcal{L}_{BCE}\left(0, f_s\left(\mathcal{F}_\theta(q), P_j^-\right)\right) + \frac{1}{|Q_u|} \sum_{q \in Q_u} \mathcal{L}_{BCE}\left(1, f_s\left(\mathcal{F}_\theta(q), P_j^-\right)\right). \tag{11}$$

Then, we utilize the NP $P_j^- \in P^-$ of each novel class to replace the input $p_c^r$ in Eq. 4, resulting in the NP $p^-$ for the entire task. Considering the diverse appearances of unknown samples (e.g., both dogs and tigers can be considered unknown for the 'cat' class), we propose employing multiple NPs for each task. Specifically, we apply $N$ MLPs to Eq. 4, yielding $N$ distinct NPs:

$$p_k^- = f_k^{mlp}\left(p_{avg}^-\right), \quad k \in \{1, \dots N\}. \tag{12}$$

## 4.3 Swap Alignment Module

Although we have generated multiple NPs for each task, not all NPs are effective, as detailed in Section 5.4. To address this, we propose the Swap Alignment Module (SA) to facilitate the generation of diverse and suitable NPs for tasks involving different classes.

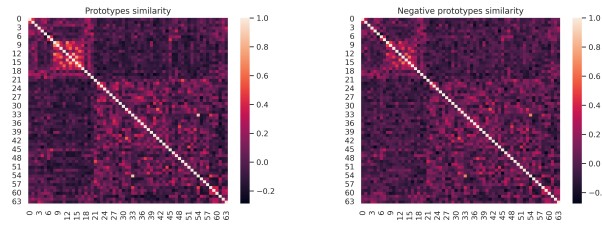

(a) Prototypes                 (b) Negative Prototypes

**Figure 5: Heatmap visualization of cosine similarity between classes. (a) shows similarity using class prototypes, and (b) displays similarity between negative prototypes learned by RPL [6]. Both approaches yield highly concordant similarity relationships.**

Inspired by ATTG [15], we employ a conjugate method to sample a pair of tasks $\mathcal{T}_1 = \left(S^1, Q_k^1, Q_u^1 \mid C_k^1\right)$ and $\mathcal{T}_2 = \left(S^2, Q_k^2, Q_u^2 \mid C_k^2\right)$, with the property that $Q_k^1 = Q_u^2$ and $Q_u^1 = Q_k^2$. This implies that the known few-shot classes $C_k^1$ in $\mathcal{T}_1$ and $C_k^2$ in $\mathcal{T}_2$ serve as negative sources for each other. We refer to $\mathcal{T}_1$ and $\mathcal{T}_2$ as each other's swap tasks.

For these swap tasks, our model generates prototypes and NPs, denoted as $[P_1, P_1^-]$ and $[P_2, P_2^-]$, respectively. We then apply the swap alignment operations :

$$\left(P_{i,a}, P_{i,a}^-\right) = \text{GCN}\left(P_i, P_i^-\right), \quad i \in \{1, 2\}, \tag{13}$$

$$\mathcal{L}_n^{\text{align}} = \sum_i \text{argmax}_j \, \textbf{sim}\left(P_{1,a}^{i,-}, P_{2,a}^j\right) + \sum_j \text{argmax}_i \, \textbf{sim}\left(P_{2,a}^j, P_{1,a}^{i,-}\right). \tag{14}$$

The GCN is a lightweight GCN network used for information propagation between prototypes and NPs, $P_{2,a}^j$ is the prototype of the $j$-th novel class in task $\mathcal{T}_2$, and $P_{1,a}^{i,-}$ is the $i$-th generated negative prototype in task $\mathcal{T}_1$. For each pair of swap tasks, we minimize the distance between each negative prototype in $\mathcal{T}_1$ and the most similar novel prototype in $\mathcal{T}_2$, and vice versa. This alignment ensures that the unknown classes in $\mathcal{T}_1$ (represented by the negative prototypes) correspond to the novel classes in $\mathcal{T}_2$ (represented by the prototypes). The alignment loss $\mathcal{L}_n^{\text{align}}$ aims to bring the NPs of task $\mathcal{T}_1$ closer to the prototypes of task $\mathcal{T}_2$, as they represent the same set of samples ($Q_u^1 = Q_k^2$). By reducing the distance between NPs and prototypes, the NPs become more discriminative and better approximate the distribution of real samples.

For task $\mathcal{T}_1$ in the swap task pair $(\mathcal{T}_1, \mathcal{T}_2)$, the final training loss function is:

$$\mathcal{L}_{\mathcal{T}_1} = \mathcal{L}_{CE}\left(Q_k^1 \cup Q_u^1\right) + \alpha \mathcal{L}_n^{neg} + \beta \mathcal{L}_n^{\text{align}} \tag{15}$$

where $\alpha$ and $\beta$ are hyperparameters. Similarly, we calculate $\mathcal{L}_{\mathcal{T}_2}$ for task $\mathcal{T}_2$. The total swap training loss is $\mathcal{L} = \mathcal{L}_{\mathcal{T}_1} + \mathcal{L}_{\mathcal{T}_2}$.

**Table 1: Closed-set ACC and open-set AUROC on two datasets. For each test, 4800 tasks are randomly sampled to ensure a confidence interval within ±0.3.**

| Algorithm | Remark | MiniImagnet | | | | TieredImageNet | | | |
|---|---|---|---|---|---|---|---|---|---|
| | | 5-way 1-shot | | 5-way 5-shot | | 5-way 1-shot | | 5-way 5-shot | |
| | | ACC | AUROC | ACC | AUROC | ACC | AUROC | ACC | AUROC |
| PEELER | CVPR2020 | 58.31±0.58 | 61.66±0.62 | 61.66±0.62 | 61.66±0.62 | - | - | - | - |
| PEELER-Res12 | CVPR2020 | 65.86±0.85 | 65.86±0.85 | 65.86±0.85 | 65.86±0.85 | 69.51±0.92 | 65.20±0.76 | 84.10±0.66 | 73.27±0.71 |
| SnaTCHer-F | CVPR2021 | 67.02±0.85 | 68.27±0.96 | 82.02±0.53 | 82.02±0.53 | 70.52±0.96 | 74.28±0.80 | 84.74±0.69 | 82.02±0.64 |
| SnaTCHer-T | CVPR2021 | 66.60±0.80 | 70.17±0.88 | 81.77±0.53 | 76.66±0.78 | 70.45±0.95 | 74.84±0.79 | 84.42±0.68 | 82.03±0.66 |
| SnaTCHer-L | CVPR2021 | 67.60±0.83 | 69.40±0.92 | 82.36±0.58 | 76.15±0.83 | 70.85±0.99 | 74.95±0.83 | 85.23±0.64 | 80.81±0.68 |
| RFDNet-Res12 | TMM2022 | 66.23±0.80 | 71.37±0.80 | 82.44±0.54 | 80.31±0.59 | 66.84±0.89 | 72.68±0.76 | 82.64±0.63 | 80.63±0.63 |
| ATT | CVPR2022 | 67.64±0.81 | 71.35±0.68 | 82.31±0.49 | 79.85±0.58 | 69.34±0.95 | 72.74±0.78 | 83.82±0.63 | 78.66±0.65 |
| ATT-G | CVPR2022 | 68.11±0.81 | 72.41±0.72 | 83.12±0.48 | 79.85±0.57 | 70.58±0.93 | 73.43±0.78 | 85.38±0.61 | 81.64±0.63 |
| GEL | CVPR2023 | 68.26±0.85 | 73.70±0.82 | 83.05±0.55 | 82.29±0.60 | 70.50±0.93 | **75.86±0.81** | 84.60±0.65 | 81.95±0.72 |
| ASOP-L | ICIP2023 | 67.85±0.20 | 71.91±0.70 | 82.81±0.30 | 81.04±0.20 | 71.49±0.40 | 75.04±0.40 | 85.15±0.10 | 81.51±0.10 |
| **DNPG** | **Ours** | **69.10±0.29** | **74.18±0.28** | **83.74±0.19** | **83.64±0.19** | **71.52±0.32** | 75.70±0.21 | **85.54±0.22** | **82.94±0.23** |

**Table 2: Closed-set ACC and open-set AUROC on CIFAR-FS dataset.**

| Algorithm | 5-way 1-shot | | 5-way 5-shot | |
|---|---|---|---|---|
| | ACC | AUROC | ACC | AUROC |
| PEELER-Res12 | 71.47±0.67 | 71.28±0.57 | 85.46±0.47 | 75.97±0.33 |
| RFDNet-Res12 | 73.83±0.92 | 75.35±0.77 | 85.12±0.74 | 84.40±0.64 |
| ATT-G | 72.43±0.65 | 76.72±0.59 | 86.52±0.49 | 84.64±0.38 |
| GEL | 76.67±0.90 | 79.43±0.72 | 87.63±0.62 | 86.84±0.58 |
| **DNPG** | **77.26±0.30** | **80.83±0.19** | **88.68±0.20** | **87.90±0.13** |

**Table 3: The 5-way 1-shot FSOR result on the COIL-DEL dataset, which is commonly used for graph classification tasks.**

| | ACC | AUROC |
|---|---|---|
| TANE-G | 75.24 | 76.70 |
| baseline | 73.20 | 73.41 |
| **DNPG (ours)** | **75.51** | **77.25** |

## 5 EXPERIMENTS

### 5.1 Datasets and Implementation Details

**Datasets.** We conduct FSOR experiments are conducted on three widely-used few-shot learning datasets: MiniImageNet [34], Tiered-ImageNet [26], and CIFAR-FS [3]. MiniImageNet is a subset of ILSVRC-12 [28], consisting of 100 categories with 600 images each. The dataset is divided into meta-training, meta-validation, and meta-testing sets with 64, 16, and 20 categories, respectively. TieredImageNet, also a subset of ILSVRC-12, contains 608 categories divided into 351, 97, and 160 categories for the respective sets. CIFAR-FS is derived from CIFAR100 [19], comprising 60,000 images across 100 categories, with the same division scheme as MiniImageNet.

**Implementation Details.** Following the settings of GEL [35] and ATTG [15], we employ ResNet-12 as the feature extractor. During the pre-training phase, we follow the protocol of Tian et al. [33] and Huang et al. [15], pre-training ResNet-12 and a linear classifier with a combination of cross-entropy loss and self-supervised rotation loss on the base set for 90 epochs. We use SGD as the optimizer, with an initial learning rate of 0.05, decayed by a factor of 10 at epoch 60. In the meta-training phase, the learning rate is set to 0.0001 for ResNet-12 and 0.05 for all other layers in the negative prototype generator. For FSOR testing and evaluation, we follow the task sampling strategy of Liu et al. [22], setting $N = 5$ and $K = 1, 5$. Each task includes 15 positive queries from each few-shot class and 5 negative classes, each with 15 negative queries. We employ cosine similarity as the similarity function, utilizing two different temperature coefficients for comparing samples with prototypes ($sim1$) and negative prototypes ($sim2$). Additionally, a trainable bias is added to $sim2$, allowing for an adjustable similarity offset for all samples to the negative prototype. Classification into the sixth class (representing the unknown class) is based on Equation 3, with the sample's probability of belonging to this class determined accordingly. The hyperparameters $\alpha$ and $\beta$ are both set to 1.

**Metrics.** Consistent with prior work [10, 15, 31, 35], we evaluate closed-set classification performance using the accuracy metric (ACC) and open-set recognition performance using the Area Under the ROC Curve (AUROC). Higher values of ACC and AUROC indicate superior performance.

### 5.2 Few-Shot Open-Set Recognition

We evaluate our model against state-of-the-art (SOTA) few-shot open-set recognition approaches, which can be categorized into two groups: Negative-Prototype-Based methods, including ATT, ATT-G [15], and ASOP-L [18], and threshold-based models, such as PEELER [22], SnaTCHer [17], RFDNet [10], and GEL [35]. Notably, PEELER and RFDNet originally employ Res-18 as the feature extractor, but Wang et al. [35] implemented Res-12 versions, denoted as PEELER* and RFDNet*, whose results we directly quote for comparison. For other models, we use the results reported in their respective papers.

Tables 1 and 2 present our model's performance compared to other approaches on three datasets. Our method surpasses all competing methods in both 1-shot and 5-shot settings across all datasets.

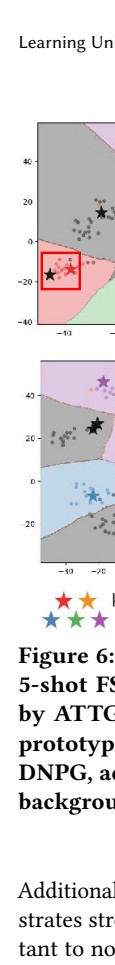
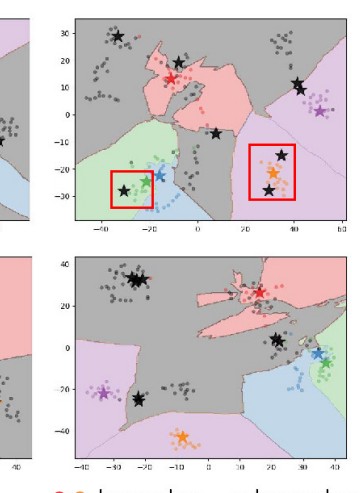

**Figure 6: Visualization of prototypes and NPs for a 5-way-5-shot FSOR task on CIFAR-FS. The first row shows NPs by ATTG, often inaccurately approximating known class prototypes (red boxes). The second row shows NPs by our DNPG, accurately representing unknown class space (gray background). Each column is a different episode.**

Additionally, on the COIL-DEL graph dataset [27], our model demonstrates strong FSOR performance, as shown in Table 3. It is important to note that the innovation of our model lies in its ability to learn unknowns from unknowns. This is highlighted by the fact that our model consistently outperforms the ATTG model, which also utilizes information from both novel and base classes during testing, on all datasets. This indicates that our approach of learning unknowns from unknowns is indeed effective for the FSOR task. Given that the ATTG model outperforms the ATT model, we select ATTG as the representative of existing Negative-Prototype-Based FSOR models in subsequent experiments.

To further demonstrate the superiority of our DNPG model, we provide visualizations of prototypes and NPs generated by both the ATTG and DNPG models in Figure 6. For both models, we set $N = 8$, meaning that 8 MLPs are used to generate 8 NPs for each task. It can be observed that the NPs generated by the ATTG model, which relies on support samples as inputs, tend to erroneously converge towards the prototypes of the corresponding classes, thereby losing their ability to effectively model unknown samples. Conversely, our DNPG model does not suffer from this issue, maintaining the distinctiveness of NPs from the prototypes.

## 5.3 Ablation Study

To assess the contribution of each component in the DNPG model, we conducted an ablation study with $N = 5$, meaning that 5 MLPs are used to generate 5 NPs for each task. In this context, MNG refers to the proposed NPs generator. Although the RPC module is not a novel contribution of our work, it is an integral part of our model; thus, for fairness, we also present results for the baseline combined with RPC. SA denotes the Swap Alignment Module. Since

the SA module is designed based on Conjugate Training (CT) [15], we include an ablation model, baseline + RPC + MNG + CT, which excludes the SA module and solely employs conjugate training.

Table 4 shows the performance of each ablation model on the MiniImageNet dataset for 5-way 1-shot and 5-way 5-shot FSOR tasks. The results demonstrate that the DNPG model significantly outperforms the baseline in both ACC and AUROC metrics. Additionally, each individual module contributes positively to the overall performance of the model.

By comparing the baseline + RPC + MNG + CT model with the baseline + RPC + MNG + SA model, it is evident that conjugate training does not markedly enhance our model. This is because conjugate training is aimed at mitigating the overfitting of generated NPs to the pseudo-unknown class samples. However, our model does not rely on pseudo-unknown class samples for generating NPs, thereby circumventing the overfitting issue. In contrast, the SA module significantly boosts the performance of our model, underscoring its importance in the DNPG framework.

**Table 4: Ablation study of our model. We report the 5-way 1-shot and 5-way 5-shot results on MiniImageNet. Our innovations are bolded.**

| Method | 5-way 1-shot | | 5-way 5-shot | |
|---|---|---|---|---|
| | Acc | AUROC | ACC | AUROC |
| baseline | 65.16±0.29 | 71.32±0.28 | 81.65±0.19 | 78.96±0.23 |
| baseline + RPC | 66.66±0.29 | 71.94±0.29 | 81.77±0.20 | 77.99±0.24 |
| baseline + RPC + **MNG** | 68.24±0.29 | 73.54±0.27 | 83.34±0.19 | 82.44±0.20 |
| baseline + RPC + **MNG** + CT | 68.22±0.29 | 73.48±0.28 | 83.33±0.19 | 82.44±0.18 |
| baseline + RPC + **MNG** + **SA** | **69.1±0.29** | **74.18±0.28** | **83.75±0.19** | **83.63±0.19** |

## 5.4 Further Analysis

**Better Negative Prototypes, Wider Unknown Space.** To further validate that our model's NPs contain minimal known class information and can model a broader unknown space, we conduct experiments using two distinct datasets to generate NPs and analyze their differences.

We train three FSOR models, ATTG [15], GEL [35], and our DNPG, on the training set of MiniImageNet. After fixing all parameters, we evaluate the models on both the test set of MiniImageNet and the test set of CIFAR-FS. We apply Principal Component Analysis (PCA) to reduce the dimensionality of the generated NPs in ATTG and DNPG and visualize them. Since GEL is not a Negative-Prototype-Based model, it does not generate NPs.

Figures 7a and 7b show the distributions of prototypes and NPs generated by the ATTG and DNPG models for both MiniImageNet and CIFAR-FS datasets. The NPs generated by the ATTG model display noticeable differences between the two datasets, indicating the incorporation of dataset-specific information. In contrast, our DNPG model produces universal NPs that effectively model a wide range of unknown spaces, making it difficult to distinguish the learned NPs between the two datasets.

Lastly, we compare the ability of different models to fit unknown class data using the learned NPs. Specifically, we calculate the maximum similarity between each unknown class sample and the NPs during testing, as depicted in Figure 8. It is evident that the NPs learned by our model exhibit superior ability to fit various

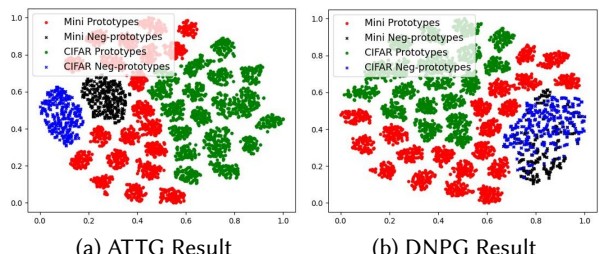

(a) ATTG Result    (b) DNPG Result

**Figure 7: (a) The ATTG model generated prototypes and NPs for both MiniImagenet and CIFAR-FS, which hibit distinct differences between the two datasets due to the incorporation of dataset-specific information, and limited to a very small part of the space. (b) Our DNPG model generated prototypes and NPs for both MiniImagenet and CIFAR-FS, which are more indistinguishable and model a larger part of the space.**

unknown class samples, further validating the effectiveness of our approach.

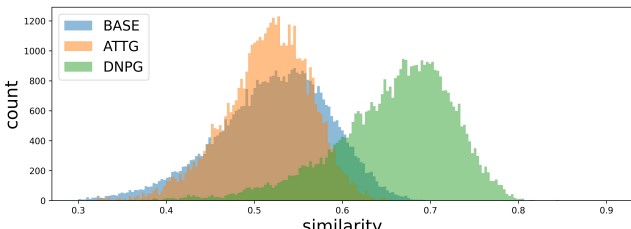

**Figure 8: The distribution of cosine similarity between the generated NPs and unknown samples, in the baseline model, ATTG model, and DNPG (ours) model. NPs generated by our model can fit unknown class samples better.**

**How does the SA Module Work?** To enhance the model's adaptability to tasks composed of diverse data, we generate multiple NPs for each task. Existing methods, such as ATTG [15], use the distance between a sample and its most similar NP as the classification score for the unknown class during both training and testing. However, simply employing multiple NPs does not ensure sufficient differentiation among them to effectively model samples from different unknown classes. In our experiments, we observed that some NPs tend to converge, collapsing to the same point in the unknown space, resulting in underutilization of the NPs.

We incrementally increase the hyperparameter $N$ (the number of generated NPs) from 1 to 8 and record the changes in model performance with (w) and without (wo) the SA module. Simultaneously, we track the number of NPs that are utilized at least once during the entire testing process. The results, depicted in Figure 9, reveal that without the SA module, the model can effectively use up to 3 NP generators. With the SA module, the model can fully utilize each generator, and the performance improves with the growth of $N$ until $N = 5$, where the best model performance is achieved.

Some models attempt to address the issue of collapsing by incorporating an equipartition constraint [25, 36], which introduces a fixed-sized feature space for prototype assignment. However, in our experiments, we found that this approach fails to resolve the collapsing problem in the FSOR task. In both the DNPG model and the model without the SA module (DNPG - SA), we applied the

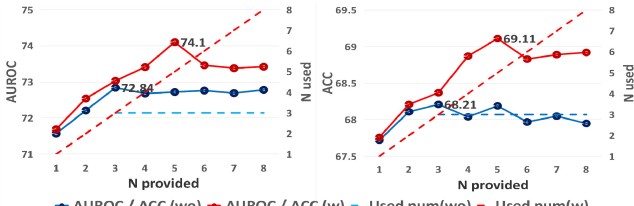

**Figure 9: As the number of generated NPs ($N$) increases. When the SA module is not used, only a maximum of 3 NPs will actually be used (blue dashed line), after using the SA module, each NP can be used (red dashed line), and the effect is further improved (the red line continues to rise).**

equipartition constraint, similar to the method outlined in [25, 36], to restrict the number of negative prototypes matching with samples. The results, as shown in Table 5 , indicate that adding the equipartition constraint negatively affects the model's performance on the FSOR task. In FSOR task, forcing each negative prototype to match in every episode is not suitable because the composition of known and unknown classes continuously varies across episodes. In certain cases, the unknown classes may consist of a similar set of classes, such as "Dalmatian" and "Alaskan Malamute." In such instances, it is more appropriate to allow the same negative prototype to represent them instead of forcing the use of a completely dissimilar negative prototype. This issue does not arise in our SA module, as we enable automatic matching between prototypes and negative prototypes, providing greater flexibility.

**Table 5: When adding the equipartition constraint (EC) to our DNPG model ( both with and without the SA module), the changes in the 5-way 1-shot ACC and AUROC on the miniImageNet dataset.**

|  | ACC | AUROC |
| --- | --- | --- |
| DNPG - SA | $67.96_{\pm0.31}$ | $72.74_{\pm0.21}$ |
| DNPG - SA + EC | $67.62_{\pm0.27}$ | $72.15_{\pm0.19}$ |
| **DNPG** | $\mathbf{69.10}_{\pm0.29}$ | $\mathbf{74.18}_{\pm0.28}$ |
| DNPG + EC | $67.70_{\pm0.32}$ | $72.40_{\pm0.21}$ |

## 5.5 Conclusion

In this paper, we investigate the task of few-shot open-set recognition. To generate diversified negative prototypes for identifying unknown class samples, we propose the DNPG model. DNPG decouples the known class sample information from the generation of negative prototypes. Instead, DNPG transfers the information learned from the unknown representation on the base classes to generate negative prototypes corresponding to novel classes. This approach ensures that the generated negative prototypes do not contain excessive known class sample information, thereby covering a broader range of unknown spaces. To ensure that each negative prototype has a certain performance and models the unknown space from different directions, we introduce the SA module, which guaranteed that multiple NPs will not collapse to the same point in the unknown space. Extensive experiments validate the effectiveness of our approach.

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
