# OpenReview forum: "Learning Unknowns from Unknowns: Diversified Negative Prototypes Generator for Few-shot Open-Set Recognition"
_acmmm.org/ACMMM/2024/Conference — MM2024 Poster_

### Official Review · Reviewer_cwae · 2024-05-23

**Rating:** 5
**Confidence:** 2

**Summary:**

This paper proposes a novel approach, termed Diversified Negative Prototypes Generator (DNPG), for Few-shot Open-set Recognition (FSOR). DNPG adopts the principle of "learning unknowns from unknowns". It leverages the unknown space information learned from base classes to generate more representative negative prototypes for novel classes. Experiments demonstrate the superior performance of the proposed method. The main contributions of this paper are:

- It proposes a novel FSOR method, the Diversified Negative Prototypes Generator (DNPG), to generate diversified negative prototypes by learning unknowns from unknowns, using the inverse representation of base classes.
- It introduces the Swap Alignment module to prevent NP collapse and enhance adaptation to novel-class data, further increasing the diversity of NPs.

**Strengths:**

- The problem studied in this paper is interesting and valuable.
- This paper provides some novel perspectives for FSOR.
- This paper is well written and in good sharp, which is easy to follow.
- The experimental results are promising.

**Limitations:**

- In Section 5.3, the authors only verify the contribution of each component on the MiniImageNet dataset. What about other datasets?
- Additionally, in Section 5.4, the authors only add the equipartition constraint (EC) to the DNPG model on the MiniImageNet dataset. What about other datasets?

**Suitability:**

2

---

### Official Review · Reviewer_Aaff · 2024-05-26

**Rating:** 4
**Confidence:** 3

**Summary:**

The paper proposes a modification of [15], a metric-based meta-learning framework for FOSR based on a negative prototype. The novel components of the proposed method include diversification of negative prototypes and a swap alignment module to sample most effective negative prototypes.

**Strengths:**

- The proposed method achieves the SOTA result.

**Limitations:**

- The proposed method is complicated, but its superiority over the current SOTA models is only marginal. Due to its complicated aspects, it is hard to know what purpose each component of the proposed method serves, and what it affects
 - Comparison to GEL may not be so accurate since its performance result was merely taken from the reference. In this aspect, the small improvement over GEL may be due to random seed difference. And in some cases, GEL outperforms the proposed method.
 - The training/inference computational cost is not discussed.
 - Does the trend in Fig. 9 hold for TieredImageNet as well? Namely, is N=5 the best result?
 - What is the final training loss, and what score is used for open-set recognition in the inference stage?

**Suitability:**

3

---

### Official Review · Reviewer_QoND · 2024-05-27

**Rating:** 4
**Confidence:** 3

**Summary:**

The paper presents a novel approach, Diversified Negative Prototypes Generator (DNPG), for Few-Shot Open-Set Recognition (FSOR), addressing the challenge of recognizing both known and unknown classes with limited labeled data. The authors propose a method that leverages the unknown space information learned from base classes to generate more representative negative prototypes for novel classes. By pre-training on the base classes to learn the unknown space representation, and utilizing inter-class relationships in the meta-learning process to construct negative prototypes for novel classes, the proposed DNPG model aims to cover a broader unknown space, achieving state-of-the-art performance on standard FSOR datasets.

**Strengths:**

1. The paper discusses the experimental validation of the model, using MiniImageNet and CIFAR-FS datasets, and evaluates the model's performance in comparison with existing methods. The authors demonstrate the effectiveness of the DNPG model in learning negative prototypes (NPs) with minimal known class information, simulating a more diverse unknown space. The model's capability to adapt to varying unknown class data is highlighted through comparative analysis with other FSOR models. Additionally, the paper details the functionality and performance enhancement brought by the introduced Swap Alignment (SA) module, ensuring adequate diversity among multiple NPs and addressing convergence issues.

2. The paper's theoretical foundation is well-developed. The use of open weights to directly generate negative prototypes from the unknown space is a unique and promising approach.

3. The paper effectively demonstrates the effectiveness of the DNPG model through comprehensive experimentation. The use of PCA for visualization and analysis of the generated negative prototypes adds depth to the experimental design.

4. The paper provides a strong comparative analysis between the DNPG model and existing methods, such as ATTG and GEL, showcasing the superiority of the proposed approach in simulating a broader unknown space and adapting to various unknown class data compositions.

5. The proposed Swap Alignment (SA) module effectively addresses the diversity and adaptability of the generated negative prototypes, contributing to a significant performance improvement. The thorough discussion and validation of this component add substantial value to the paper.

**Limitations:**

1. While the experimental results on MiniImageNet and CIFAR-FS datasets showcase the effectiveness of the DNPG model, the paper could benefit from validation on additional datasets to demonstrate its generalizability across diverse data distributions and characteristics.

2. The paper lacks a detailed error analysis, which could provide insights into specific failure cases and the model's limitations in certain scenarios. A more in-depth exploration of model errors would contribute to a nuanced understanding of the DNPG model's strengths and weaknesses.

3. The paper does not extensively discuss the computational complexity and efficiency of the proposed approach, including training time, memory usage, and inference speed. Providing insights into the computational requirements would offer a more comprehensive evaluation of the practical viability of the DNPG model.

4. The paper does not delve deeply into the sensitivity of the proposed DNPG model to its hyperparameters. A comprehensive analysis of the model's performance across a range of hyperparameter settings would enhance the understanding of its robustness and stability under different configurations.

**Suitability:**

2

---

### Official Review · Reviewer_iBmt · 2024-05-27

**Rating:** 4
**Confidence:** 3

**Summary:**

This paper delineates a novel approach (termed DNPG) to Few-Shot Open-Set Recognition (FSOR), where a model needs to recognize known classes and identify unknown classes with limited labeled data. The paper claims that existing approaches mainly generate negative prototypes based on known class data, which limits their representation capability. In addressing this issue, the DNPG leverages the unknown space information learned from base classes to create more representative negative prototypes for novel classes.

**Strengths:**

S1. The paper's significance is underscored by the proposed DNPG, which exhibits an amalgamation of the diversified multiple negative prototype generator and swap alignment.

S2. The clarity of the manuscript enhances accessibility for readers, facilitating a straightforward understanding of the proposed approach.

S3. The evaluation is comprehensive, with comparisons to baseline methods providing a compelling demonstration of the effectiveness of the DNPG model.

**Limitations:**

W1. The manuscript could explicate the GCN in the SA module, a factor which could be pivotal for understanding the proposed SA, e.g., its motivation and interpretation.

W2. The computational complexity of the DNPG model, particularly the attention mechanism, which could be a concern for large-scale datasets, is not discussed in the manuscript.

W3. The paper could further explain and analyze the results in Figure 5.

Typos:

- Line 284: fynction

- $d$ in Eq. (7).

**Suitability:**

1

---

### Meta-Review · Area_Chair_v5Kb · 2024-07-02

**Recommendation:** Accept (Poster)
**Confidence:** 5

**Metareview:**

This work proposes the Diversified Negative Prototypes Generator (DNPG), to generate diversified negative prototypes using open weights from the unknown space while previous methods usually generate negative prototypes based on known class data. The idea is novel and sound, with the potential to attract interest from the open-set community. All the reviewers are satisfied with the author rebuttal. Despite that, the authors are encouraged to incorporate the reviewers' feedbacks into the revised version, and provide detailed computational complexity of their proposed model and the new components.